# Prioritization of the concepts and skills in quantitative education for graduate students in biomedical science

**Louis J. Gross** [1,2]*, **Rachel Patton McCord** [3,4], **Sondra LoRe** [5¤a], **Vitaly V. Ganusov** [1,4,6,7], **Tian Hong** [1,3,4], **W. Christopher Strickland** [1,2,7], **David Talmy** [1,4,6], **Albrecht G. von Arnim** [3,4], **Greg Wiggins** [1¤b]

**1** National Institute for Mathematical and Biological Synthesis, University of Tennessee, Knoxville, TN, United States America, **2** Department of Ecology and Evolutionary Biology, University of Tennessee, Knoxville, TN, United States America, **3** Department of Biochemistry & Cellular and Molecular Biology, University of Tennessee, Knoxville, TN, United States America, **4** UT-ORNL Graduate School of Genome Science and Technology, University of Tennessee, Knoxville, TN, United States America, **5** National Institute for STEM Evaluation and Research, University of Tennessee, Knoxville, TN, United States America, **6** Department of Microbiology, University of Tennessee, Knoxville, TN, United States America, **7** Department of Mathematics, University of Tennessee, Knoxville, TN, United States America

¤a Current address: Center for Academic Research & Excellence (CARE), Chattanooga State Community College, Chattanooga, TN, United States of America
¤b Current address: NC Dept. of Agriculture and Consumer Services, Plant Industry Division, Cary, NC, United States of America
* lgross@utk.edu

**Data Availability Statement:** Data in CSV format are uploaded as supplemental information.

**Funding:** Burroughs Wellcome Fund Quantitative and Statistical Thinking in the Life Sciences Award

## Abstract

Substantial guidance is available on undergraduate quantitative training for biologists, including reports focused on biomedical science. Far less attention has been paid to the graduate curriculum and the particular challenges of the diversity of specialization within the life sciences. We propose an innovative approach to quantitative education that goes beyond recommendations of a course or set of courses or activities, derived from analysis of the expectations for students in particular programs. Due to the plethora of quantitative methods, it is infeasible to expect that biomedical PhD students can be exposed to more than a minority of the quantitative concepts and techniques employed in modern biology. We collected key recent papers suggested by the faculty in biomedical science programs, chosen to include important scientific contributions that the faculty consider appropriate for all students in the program to be able to read with confidence. The quantitative concepts and methods inherent in these papers were then analyzed and categorized to provide a rational basis for prioritization of those concepts to be emphasized in the education program. This novel approach to prioritization of quantitative skills and concepts provides an effective method to drive curricular focus based upon program-specific faculty input for science programs of all types. The results of our particular application to biomedical science training highlight the disconnect between typical undergraduate quantitative education for life science students, focused on continuous mathematics, and the concepts and skills in graphics, statistics, and discrete mathematics that arise from priorities established by biomedical science faculty. There was little reference in the key recent papers chosen by

#1018963 to the University of Tennessee and the National Science Foundation through Award DBI-1300426 to the University of Tennessee.

**Competing interests:** The authors have declared that no competing interests exist.

faculty to classic mathematical areas such as calculus which make up a large component of the formal undergraduate mathematics training of graduate students in biomedical areas.

## Introduction

There is widespread agreement that training in quantitative approaches is critical for life science students at the undergraduate level [1–3] and for graduate students in biomedical fields [2, 4]. Less clarity exists about how to prioritize which quantitative concepts and approaches are most important for students to learn. This is a central concern in all of education: deciding what components to include, when to do so, how to reinforce concepts and skills deemed of high importance, and how to evaluate the success of a curriculum through analysis of student learning outcomes. Typical approaches to this challenge are to base curricular decisions on: history (e.g. the standard pre-college collection of topics in arithmetic, geometry, algebra and trigonometry); reports of authoritative groups in professional societies and committees supported by federal agencies and foundations through organizations such as the National Academies of Science, Engineering and Medicine [1, 2, 5] and requirements for accreditation [6]. There are hosts of different quantitative topics, conceptual approaches, and skills, and so training typically involves trade-offs on the concepts taught. It is within this context that local decisions by program faculty moderate curricular choices. At advanced levels, the availability of expertise among the instructors in a particular academic program may introduce constraints as well, particularly if at-distance and on-line options are not included.

In general, there has been more explicit guidance and study of how to incorporate quantitative skills into undergraduate biology programs [1, 3, 7–11] than there has been at the graduate level. Along with general guidance for graduate STEM education [12], studies have emphasized the differences in graduate quantitative training between medical programs and mathematical biology [13] and suggestions for integrating particular quantitative topics in biology graduate education [14, 15]. A recent workshop provided recommendations on quantitative biology education for life science graduate programs [16]. The report noted that the breadth of quantitative concepts requires flexibility of quantitative education initiatives and encouraged methods to foster student-centered learning. A recommendation was that guidance be developed to prioritize quantitative concepts and skills as (i) essential for all students in a life science graduate program, (ii) beneficial but not essential for all students, and (iii) helpful for some set of students in a particular program.

Aside from expert reports [2, 4] and accreditation standards [6], previous efforts to identify key quantitative concepts for graduate students in a field have typically taken two forms: (i) automated text mining of publications, or (ii) faculty deliberation and decision-making committees. For example, large samples of articles from field-specific journals have been mined for pre-defined statistical terms to inform which concepts are key to graduate training in higher education research [17, 18], ecology [19] and oncology [20]. In the second approach, there are examples from geoscience departments [21], life science programs [22], and business schools [23] which used faculty meetings or surveys of faculty, graduates, or employers to explicitly define key quantitative skills deemed essential for students in these fields.

In order to address recommendations for prioritization of quantitative concepts and skills [16], we propose a novel approach to guide the enhancement of quantitative components of the PhD curriculum. Our approach considers the differences across the diversity of graduate programs in life sciences and suggests that it is the faculty for a particular program who can most appropriately quantitative expectations based on their local needs. As an example of how

to assist programs identify concepts and skills appropriate for their students, we first asked faculty from three programs that train biomedical science graduate students to identify recent papers that are key to understanding their field. We did not ask the faculty to focus on papers with quantitative content, but rather asked them to suggest papers with key scientific content appropriate so they have an expectation of comprehension of these papers for all students in their program. We then mined the papers through expert coding to obtain the quantitative approaches represented. From this, we determined which quantitative topics occurred most frequently, providing a rational basis for prioritizing across the host of potential quantitative topics those that are closely tied to comprehension of recent scientific papers that the local faculty consider key.

Our approach gains the benefit of both broad-based faculty input and supervised data filtering. These data were analyzed and compared to information on the prior educational background and syllabi of courses taken by students in these programs. Although applied here to quantitative topics in biomedical science graduate programs, the methodology we develop is broadly applicable to any graduate education program and can provide further evidence beyond historical expectations or those derived from decision-making committees that the prioritization of topics chosen for that program is consistent with those required to read with comprehension the literature in the area.

## Materials and methods

### Data collection methodology

For the first phase, qualitative data was collected in the form of document analysis of research articles from the faculty associated with the major programs at the University of Tennessee, Knoxville (UTK) which educate graduate students in biomedical science—the Departments of Biochemistry & Cellular and Molecular Biology (BCMB), Microbiology (Micro), and the University of Tennessee-Oak Ridge National Lab (UT-ORNL) Graduate School of Genome Science & Technology (GST). During the semester of Fall 2018, faculty associated with these programs were asked to provide a single journal article published in the previous five years which they considered important for all the students in their graduate program, not just those associated with their lab, to be able to read with comprehension. These articles may be ones used in their courses and seminars, but this was not emphasized. Faculty were asked not to submit review articles and were not told to emphasize quantitative topics in the papers suggested, but to submit papers with important scientific content. Solicitations were conducted over seven weeks, resulting in 48 papers submitted from 40 respondents (S1 Table in S1 Appendix). The faculty came from three core graduate programs: BCMB, Micro, and GST and a few faculty with main appointments in additional units (Biomedical and Diagnostics, Nutrition, UT Medical Center, Plant Sciences, Biosystems Engineering and Soil Science, Animal Science, and Electrical Engineering and Computer Science) were also involved because of their affiliation with the core programs (S1 Table in S1 Appendix).

The research methods employed an exploratory sequential mixed methods design [24, 25] where qualitative methods are followed by quantitative measures (Fig 1). After the collection of articles, the six faculty on the project were asked to identify quantitative skills from a sampling of eight randomly assigned papers, spending 10–15 minutes identifying the quantitative tools/skills and/or concepts used in each article. S1 Table in S1 Appendix provides the identifying number of the faculty carrying out the initial assessment for each paper. Meeting facilitators and the evaluator, acting as a participant-observer [26, 27], kept ethnographic memos [28] regarding quantitative concepts and skills discussed. The initial review of article samples concluded with a listing of 173 quantitative skills under 21 general concepts. After consideration

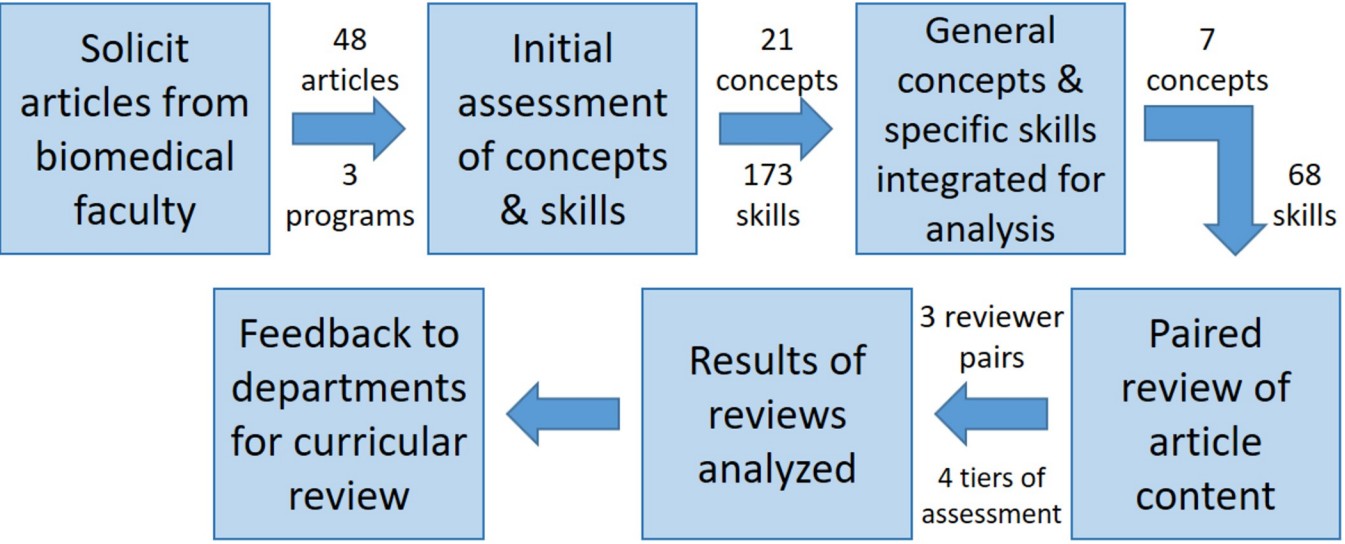

**Fig 1. Data collection and analysis.** Process for data collection and analysis of articles chosen by faculty as appropriate for all biomedical students completing a PhD in their program to read with comprehension. See main text for details of the steps taken for collection of data and analysis.

of overlaps in the quantitative topics and general concepts identified by the initial assessment, this list was distilled by the full group of faculty on the project to an aggregated listing of seven general concepts and 68 specific skills associated with the general concepts (see S2 Table in S1 Appendix for the listing of these concepts and skills).

The quantitative phase of the mixed methods research design included the creation of a survey to be used to re-analyze the submitted papers. This reanalysis was done by grouping the 48 papers with teams of two members of the faculty team assigned to consider in detail each of a set of papers that were deemed to be most connected to their backgrounds. S1 Table in S1 Appendix provides the identifier of the members of the project faculty pair carrying out this reanalysis for each paper. This second analysis of the journal article's quantitative content included four tiers of related assignments for each concept/skill: 1) the presence of generalized concepts in the sample paper, 2) the level of importance of this concept to understanding the paper, followed by 3) specific skills related to the general concept and 4) the level of importance of the specific skill to understanding the paper. The rankings for each concept or skill assessed in these tiers was: 1: not present, 2: marginally important to comprehension of the paper, 3: somewhat important to comprehension of the article, or 4: very important to comprehension of the paper (Fig 2).

This second analysis was carried out by teams of two faculty members, who each provided their own ranking of importance of the concepts and skills. After each review pair completed the surveys for their journal articles the evaluator compared the results for each article. All of the article scores were averaged collectively by concepts and specific skills. Graphics were constructed to show the existence and importance of the concepts and then presented to the research team.

## Statistical data analysis

To visualize the fractions of importance levels (1: 'not present', 2: 'marginally important', 3: 'somewhat important', and 4: 'very important') for all seven general concepts, first the overall fractions of the four levels across all concepts were calculated (S1 Fig in S1 Appendix, dashed lines). These overall fractions are defined as 'expected' fractions. Next, the fractions of the four

levels for the seven individual general concepts were calculated (Fig 2 and S1 Fig in S1 Appendix, grey bars), and plotted to visualize the deviation of the individual fractions from the expected fraction (S1 Fig in S1 Appendix). For example, the fraction of 'not present' level for the 'Modeling' concept was higher than expected by 0.31, and the fraction of 'very important' level for the same concept was lower than expected by 0.19. For each general concept, all 48 papers (for which there were 96 total evaluations since two project team members assessed each paper) were included for calculating the fractions. A chi-squared test was performed for each concept with the frequencies of the four levels and the expected frequencies derived from the overall fractions (levels were treated as categorical rather than ordinal data). This test allowed comparisons of the importance levels assigned for each general concept to the overall assessment of importance across all quantitative concepts, with a graphical perspective provided in S1 Fig in S1 Appendix.

A similar method was used to visualize the importance levels of skills within each general concept. For each general concept first the overall (i.e. expected) fractions of the four levels among the evaluations in which the concept is present were calculated (Fig 3 grey bars). Then the fractions of the four levels for individual skills within the concept were calculated. A heatmap was used for each concept to visualize the deviation of the individual fractions (per skill per level) from the expected ones (Fig 3). The fraction per skill per level was with respect to evaluations in which the concept is present, except for a few instances (<3% among the 96 evaluations) where the skill levels were missing and the fractions were with respect to available evaluations. In the heatmap, white color means expected fraction, red color means higher-than-expected fraction, and blue color means lower-than-expected fraction. A more detailed comparison of the importance values for the general concepts in Fig 3A is in S1 Fig of S1 Appendix.

## Power analyses

To evaluate how many papers contain a given number of concepts or skills power analyses were performed. For the analysis of general concepts, all evaluations were resampled with

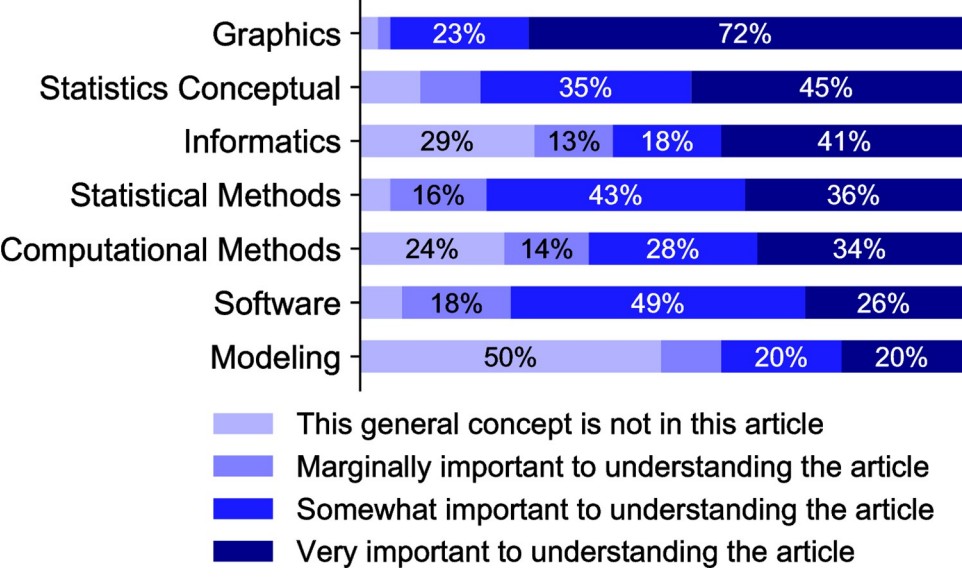

**Fig 2. General concepts.** The seven general concepts identified in the submitted papers and the proportions of importance level of each concept. Percentage values indicate the percent of papers in which a given concept was assigned a given importance level. Values greater than 10% are labeled with text.

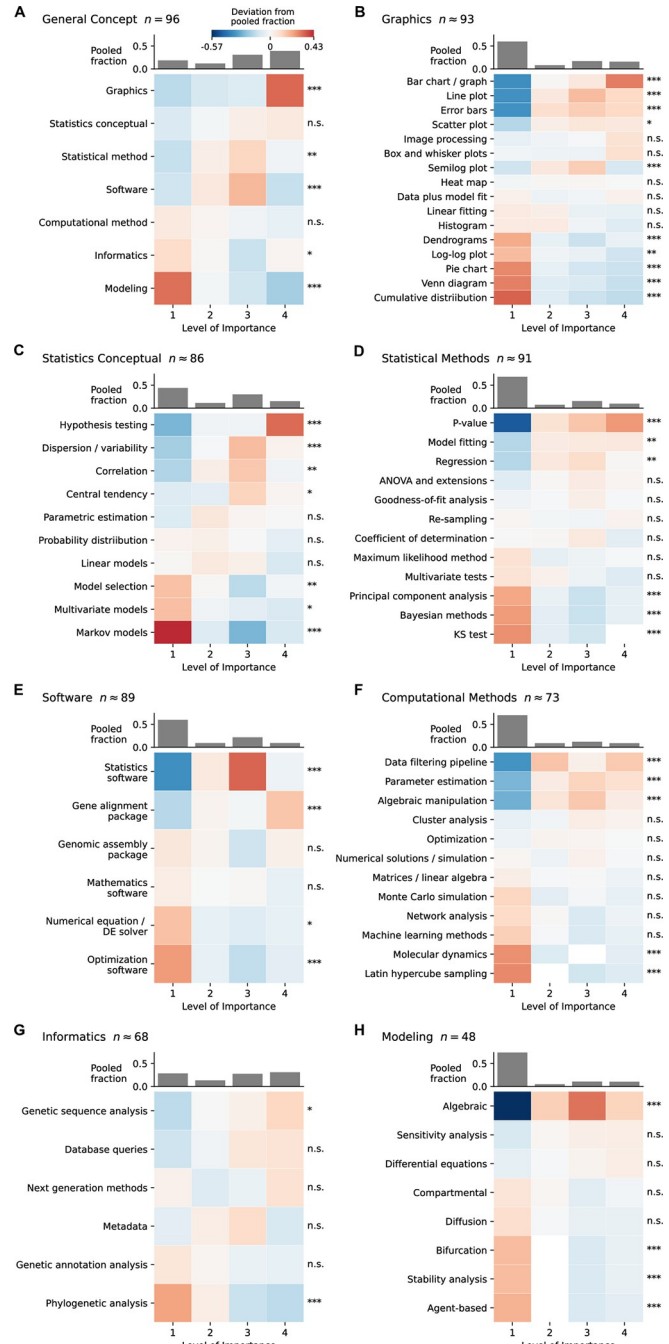

**Fig 3. Deviations of concepts and skills.** Deviations from average importance distributions for concepts (A) and skills (B to H) identified from solicited papers. Importance ranges from 1 = Concept not in this article to 4 = Very important to understanding this article, as in Fig 2. Significant deviation from expected fraction is as follows: n.s. $p > 0.05$; * $p \leq 0.05$; ** $p \leq 0.01$; *** $p \leq 0.001$. Further detail about the data is provided in the supplementary material.

replacement 1,000 times and the percent of times a given concept (out of a set of seven) was present in the sample at a particular level (defined by a cut-off) was counted. For example, a cut-off of 3 implied that this specific concept was somewhat important to understand the paper. Analysis suggests that 5 papers already have most of the concepts represented (S2A Fig

in S1 Appendix). A similar power analysis for the skills when resampling all evaluations suggests that different numbers of papers were needed to cover skills in different concepts (S2B Fig in S1 Appendix). For example, Informatics and Statistics: Conceptual skills were most contained in most papers while Computational Methods and Modeling required a larger sample size (20–25) to have their skills well represented. This analysis provides evidence that there was a sufficient sample size of articles to identify most of the defined skills, which perhaps is not surprising given that these concepts and skills were mined from the particular papers analyzed.

## Results

This case study focuses on the major units at the University of Tennessee, Knoxville (UTK) which educate PhD students in biomedical science: the Departments of Biochemistry & Cellular and Molecular Biology (BCMB), Microbiology (Micro), the University of Tennessee- Oak Ridge National Lab (UT-ORNL) Graduate School of Genome Science & Technology (GST). The three graduate programs at the core of the analysis carried out here host a total of ~160 graduate students, predominantly in the PhD track. Microbiology includes two main areas: environmental microbiology and microbial pathogenesis. BCMB encompasses mainly three areas: physical biochemistry, and molecular and cellular biology of both plant and animal systems. GST is an intercollegiate life science program that emphasizes training across the interface of the wet-lab and dry-lab, e.g., in computational biology [14]. It includes faculty members from Microbiology, BCMB, and other departments across the university.

Faculty associated with these units were asked to each identify at least one recently published journal article, not necessarily with quantitative emphasis, that they suggest biomedical science students completing a PhD in their program should be able to read with comprehension (Fig 1). The project team then analyzed these papers for quantitative concepts and methods, grouping these in a hierarchical manner based on a few core concepts, approaches, and skills necessary to adequately understand faculty-selected papers (see Materials and Methods for the full design description and S2 Table in S1 Appendix for the concepts and skills).

We obtained the distributions of the evaluations over the four importance levels (Fig 2 and S1 Fig in S1 Appendix) and found that five out of the seven general concepts showed significant deviations from the pooled distributions across concepts based on the chi-squared tests. This provides evidence that some general quantitative concepts are more important than others for biomedical graduate students in the UT programs as judged by the faculty in those programs. Interestingly, among the seven general concepts, 'Graphics' was ranked at the top in terms of the median importance levels, whereas 'Modeling' was considered the least important concept as it was absent from most of the articles (Fig 2 and S1 Fig in S1 Appendix). In addition, 'Software' and 'Statistical Methods' showed a significant deviation because they appeared in a large number of articles with a medium level of importance. Within individual general concepts, 54% of the specific skills associated with that concept showed significant deviations in importance from the pooled distributions across all skills for that concept (Fig 3). Among skills that were considered most important, 'Bar chart/graph', 'Line plot', 'Error bars', 'Hypothesis testing' and 'P-value' were not only present in most of the articles that contain the corresponding general concepts, but also scored as 'very important' in those articles. Some skills were considered significantly important even though their general concepts did not rank high, including 'Statistics software', 'Gene alignment package', 'Gene sequence alignment', 'Data filtering pipeline', 'Algebraic manipulation' and 'Parameter estimation.' Most of these skills showed medium levels of importance in the articles where the skills were present.

Having identified which concepts and skills were noteworthy in the papers important to faculty training biomedical graduate students, we considered how these skills align with typical

training of incoming graduate students in these programs. We are unaware of any formal analysis of the quantitative background of students entering US life science graduate programs. Informal evidence from students in our life science departments indicates that there is considerable variability both within programs and between programs of the formal mathematics, statistics, and computing backgrounds of these students. Curriculum requirements for undergraduate biology programs, including that at UTK, typically include calculus, but not necessarily statistics and/or data analysis/visualization courses or any specific requirements in computational biology [29, 30]. This suggests that many of the mathematical skills needed to engage with topics deemed important in our current study will need to be built during the graduate biomedical training process.

## Discussion

This study developed a novel and generalizable data collection methodology targeting faculty opinions about core concepts for their PhD students in the biomedical sciences. Upon analyzing the faculty-supplied journal articles for their quantitative, data-analytical techniques our results suggest that there is significant variability across both general concepts and particular skills with regard to their estimated importance in comprehending key papers. It is important to note that scoring of the articles was primarily based on a standard of literacy; that is, the goal was to determine the subject-area knowledge necessary to *comprehend* the material in question. An independent group of researchers who are not specific, subject-matter experts in a paper's field of study will frequently have difficulty determining the technical knowledge necessary to reproduce the study. Consequently, determining the quantitative, data-analytical techniques necessary for reproduction was beyond the scope of our work. Our comprehension-based approach has the side-effect of emphasizing quantitative techniques that have an intrinsic communication or presentation aspect (e.g., 'Graphics') at the expense of other categories (e.g., 'Computational Methods') which are more fully methods- or analysis-based. An additional limitation to our comprehension-based approach is that it is inherently biased toward subject matter whose prerequisite knowledge is ubiquitous among current and past biomedical researchers and is therefore a measure of the status-quo for biomedical research literacy rather than an indicator of field direction. More complex methods or methods requiring a high degree of uncommon background knowledge are less likely to be represented here, regardless of their merits for advancing biomedical research.

Our methodology reveals critical details about the baseline of knowledge necessary for those pursuing a PhD in the biomedical sciences today. It also provides a field-specific, detailed usage comparison of quantitative tools: bar and line plots, hypothesis-testing, statistical software packages, p-values, algebraic expressions, and data analysis software all stand out as highlights of research methodology while dynamic modeling and more advanced data-science techniques are more uncommon. The results indicate a potential disconnect between the typical formal quantitative training included in an undergraduate life science curriculum, focused on calculus and basic statistics, and the focal concepts identified in our analysis on visualization, statistical and computational methods. Understanding and quantifying this disconnect can help to inform what quantitative concepts and skills should be emphasized in the graduate biomedical curriculum.

Our analysis suggests the types of skills that should either be already present in UTK biomedical graduate students at admission or that need to be learned in graduate school. These results have strong implications for curricula of STEM-training undergraduate programs, particularly if they prove generalizable to other graduate programs training students in biomedicine. An informal survey of current biomedical graduate students at UTK indicated that the

vast majority had undergraduate courses in basic calculus and introductory statistics, with few having formal quantitative training, including in computer science or data science, beyond this. The current strong focus on continuous mathematics (e.g., basic calculus) may need to be replaced with other mathematics-based courses that put stronger emphasis on data analysis and interpretation. Furthermore, biomedical graduate programs may need to start offering courses in data science to align training with expectations of the faculty, though there is evidence that undergraduate data science programs are growing in influence [5, 31].

We would welcome further studies examining whether our results are a part of a general pattern for life science students or an outlier among the general expectations of core concepts and skills that graduate students in biomedical sciences must have. Our expectation is that if a similar analysis were undertaken in different life science disciplines such as ecology, physiology or evolution, the strong focus we identified on non-continuous mathematics for the biomedical science papers evaluated may well not arise. The long history of theory based on calculus and dynamical systems in these other biological sciences disciplines could well produce different quantitative emphases in analysis of key recent papers. The methodology we developed could provide evidence for a rational basis for inclusion of particular topics and skills in the quantitative curriculum in alternative life science disciplines. Regarding recommendations within a particular life science discipline across institutions, a possibility is for professional societies of sub-disciplines to encourage a variety of the main academic departments from several universities to carry out a similar analysis to that described here. A summary of results from these institutions could then form a basis for guidance from the professional society regarding prioritization of quantitative concepts and skills for that particular sub-discipline.

Based on comments in general reports on life science quantitative education [7–10] and those for graduate programs [16], our results provide further evidence that the diversity of quantitative skills and concepts are not fully encompassed in training of students prior to graduate work. As our analysis indicates, the scientific literature appropriate for students in a program to be exposed can encourage students to appreciate the necessity of enhancing their quantitative understanding. From our power analysis, exposure to even a relatively small number (20–30) of articles will incorporate the vast majority of quantitative topics arising as important for a program. Encouraging students to enhance their backgrounds through carrying out peer reviews has been posited as an effective means to further their career development [32] but it just as readily could foster further investigation of quantitative concepts to enhance the quality of their reviewing.

Our analysis suggests the types of skills that should be acquired during the course of biological/biomedical graduate work if not already developed at the undergraduate level. Our effort does not provide suggestions for either timing or implementation of curricular innovations to enhance the identified skills and concepts. Recommendations for such a detailed implementation include those for both biomathematics and medical school programs [13] and those targeted specifically on biological data science [31]. Building on these suggestions, we envision that such changes can be implemented as a three-tiered sequence of structured training elements, delivered through a mixture of modalities including formal courses, lab group tutorials, boot-camps, and individual on-line learning. For entry-level PhD trainees we suggest raising 'Awareness' through formal learning units to be utilized in entry-level bioscience graduate courses illustrating past (and current) trends in quantitative life science. For mid-level trainees, we suggest creating 'Keys to Success' through short and intensive training vehicles that build competence for actionable, quantitative skills based on carefully chosen experimental designs and biological data. Finally, fostering a self-sustaining 'Peer-Learning Community' by networking more advanced students (including those with more extensive prior quantitative

training) with junior trainees in the form of informal tutorials, user groups, or journal clubs that collaboratively review preprints [32] could sharpen particular skills. Together, these training elements could change the mindset of biomedical PhD students to encourage their expenditure of the effort needed to utilize particular quantitative skills as well as their appreciation that an integrated collection of quantitative skills is fundamental to their career success.

## Supporting information

**S1 Appendix. Supplementary materials with S1, S2 Figs, S1 and S2 Tables.**
(DOCX)

**S1 Dataset. Coded results scaled from 1 (not present) to 4 (very important) for all analyzed papers and associated concepts and skills.**
(CSV)

**S1 File.**
(CSV)

## Acknowledgments

The authors thank Louis Becker and UT Libraries for assistance with literature review.

## Author Contributions

**Conceptualization:** Louis J. Gross, Sondra LoRe, Greg Wiggins.

**Data curation:** Louis J. Gross, Rachel Patton McCord, Sondra LoRe, Vitaly V. Ganusov, Tian Hong, W. Christopher Strickland, Albrecht G. von Arnim, Greg Wiggins.

**Formal analysis:** Louis J. Gross, Sondra LoRe, Vitaly V. Ganusov, Tian Hong, W. Christopher Strickland, David Talmy, Albrecht G. von Arnim, Greg Wiggins.

**Funding acquisition:** Louis J. Gross.

**Investigation:** Louis J. Gross, Rachel Patton McCord, Sondra LoRe, Tian Hong, W. Christopher Strickland, David Talmy, Greg Wiggins.

**Methodology:** Louis J. Gross, Sondra LoRe, Tian Hong, Albrecht G. von Arnim, Greg Wiggins.

**Project administration:** Louis J. Gross, Sondra LoRe, Greg Wiggins.

**Supervision:** Louis J. Gross.

**Visualization:** Louis J. Gross, Vitaly V. Ganusov, Tian Hong.

**Writing – original draft:** Louis J. Gross, Greg Wiggins.

**Writing – review & editing:** Louis J. Gross, Rachel Patton McCord, Sondra LoRe, Vitaly V. Ganusov, Tian Hong, W. Christopher Strickland, David Talmy, Albrecht G. von Arnim, Greg Wiggins.

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
