## [Decision Letter · Decision Letter 0]

10 Feb 2023

PONE-D-22-21390Prioritization of the concepts and skills in quantitative education for graduate students in biomedical sciencePLOS ONE

Dear Prof. Louis Joseph Gross, 

Thank you for submitting your manuscript to PLOS ONE. After careful consideration, we feel that it has merit but does not fully meet PLOS ONE’s publication criteria as it currently stands. Therefore, we invite you to submit a revised version of the manuscript that addresses the points raised during the review process.

Please submit your revised manuscript by Mach, 10 If you will need more time than this to complete your revisions, please reply to this message or contact the journal office at plosone@plos.org. Please include the following items when submitting your revised manuscript:A rebuttal letter that responds to each point raised by the academic editor and reviewer(s). You should upload this letter as a separate file labeled 'Response to Reviewers'.A marked-up copy of your manuscript that highlights changes made to the original version. You should upload this as a separate file labeled 'Revised Manuscript with Track Changes'.An unmarked version of your revised paper without tracked changes. You should upload this as a separate file labeled 'Manuscript'.If applicable, we recommend that you deposit your laboratory protocols in protocols.io to enhance the reproducibility of your results. Protocols.io assigns your protocol its own identifier (DOI) so that it can be cited independently in the future. For instructions see: https://journals.plos.org/plosone/s/submission-guidelines#loc-laboratory-protocols. Additionally, PLOS ONE offers an option for publishing peer-reviewed Lab Protocol articles, which describe protocols hosted on protocols.io. Read more information on sharing protocols at https://plos.org/protocols?utm_medium=editorial-email&utm_source=authorletters&utm_campaign=protocols.

We look forward to receiving your revised manuscript.

Kind regards,

Yaser Mohammed Al-Worafi

Academic Editor

PLOS ONE

Journal Requirements:

“The authors appreciate support from the Burroughs Wellcome Fund Quantitative and Statistical Thinking in the Life Sciences Award #1018963 to the University of Tennessee and the National Science Foundation through Award DBI-1300426 to the University of Tennessee.  The authors thank Louis Becker and UT Libraries for assistance with literature review”

“Burroughs Wellcome Fund Quantitative and Statistical Thinking in the Life Sciences Award #1018963 to the University of Tennessee and the National Science Foundation through Award DBI-1300426 to the University of Tennessee.  “

Reviewers' comments:

Reviewer's Responses to Questions

**Comments to the Author**

1. Is the manuscript technically sound, and do the data support the conclusions?

Reviewer #1: Yes

Reviewer #2: No

2. Has the statistical analysis been performed appropriately and rigorously? 

Reviewer #1: Yes

Reviewer #2: No

3. Have the authors made all data underlying the findings in their manuscript fully available?

Reviewer #1: Yes

Reviewer #2: Yes

4. Is the manuscript presented in an intelligible fashion and written in standard English?

Reviewer #1: Yes

Reviewer #2: No

5. Review Comments to the Author

Reviewer #1: Summary:

The article entitled Prioritization of the concepts and skills in quantitative education for graduate students in biomedical science elevates a long-standing question of aligning graduate student preparedness with program-based curricular. This work situates the current/historical perspectives of how program curriculum may vary by school, and department levels, and the implications of graduate student readiness and/or requirement for tailored curriculum. The novel approach of using faculty survey of literature and identifying preferential requisites necessary for graduate students in their prospective fields offers and opportunity for best practices in curriculum design in the biomedical sciences. Overall and very nice article worthy of publication.

Minor comments:

Methodology and Statistics: The methodology implemented was well-designed and sample size and statistics used were of strong rigor.

Results: Results are well present allowing for clear interpretation of author’s findings including supplemental data shown.

Minor concern: The reviewer would appreciate further discussion as it relates to how alignment of student needs locally can be expanded to discipline across institutions.

Reviewer #2: The idea is not clear

The sample size was 160 and no details on how it was obtained or tge technique used

The concepts arenot clear

Chi2 was mentioned in tge analysis but not clear in the results

All results are figures with no tables

No clear scaling system was used to quantify the data as likert scale

Most of the references used were old

6. PLOS authors have the option to publish the peer review history of their article (what does this mean?). If published, this will include your full peer review and any attached files.

Reviewer #1: No

Reviewer #2: **Yes: **Ghada Omer Hamad Abd El-Raheem

---

## [Author Response · Author response to Decision Letter 0]

26 Mar 2023

The comments below are included in the Response to Reviewers letter - that letter is duplicated here. 

We thank the editor and reviewers for helpful comments that allowed us to enhance the submission. We here respond to each review comment, using red to denote our response. 

We have updated the manuscript to use the PLOS ONE formatting guidelines.

As noted by the editor, we have removed all acknowledgement of financial support from the Acknowledgements section of the manuscript. The statement that is already in the Funding Statement of the online submission form is appropriate and need not be changed. 

Reviewer #1: Summary:

The article entitled Prioritization of the concepts and skills in quantitative education for graduate students in biomedical science elevates a long-standing question of aligning graduate student preparedness with program-based curricular. This work situates the current/historical perspectives of how program curriculum may vary by school, and department levels, and the implications of graduate student readiness and/or requirement for tailored curriculum. The novel approach of using faculty survey of literature and identifying preferential requisites necessary for graduate students in their prospective fields offers and opportunity for best practices in curriculum design in the biomedical sciences. Overall and very nice article worthy of publication.

Minor comments:

Methodology and Statistics: The methodology implemented was well-designed and sample size and statistics used were of strong rigor.

Results: Results are well present allowing for clear interpretation of author’s findings including supplemental data shown.

Minor concern: The reviewer would appreciate further discussion as it relates to how alignment of student needs locally can be expanded to discipline across institutions.

We appreciate the kind comments of Reviewer #1 and their thoughtful summary of the main objective of the paper. We appreciate their suggestion as to how local needs might be expanded to recommendations across the discipline. Our manuscript had discussed expansion of the approach we suggested to different areas of life science outside biomedical areas. Based on the reviewer’s comment though, we have added a couple of sentences to that paragraph in the Conclusions with a suggestion that a professional society for a particular life science sub-discipline might encourage some of the training programs at different institutions to carry out an analysis similar to ours, and then use these to provide overall guidance on quantitative concepts for their sub-discipline. 

1. Is the manuscript technically sound, and do the data support the conclusions?

Reviewer #1: Yes

Reviewer #2: No

2. Has the statistical analysis been performed appropriately and rigorously?

Reviewer #1: Yes

Reviewer #2: No

3. Have the authors made all data underlying the findings in their manuscript fully available?

Reviewer #1: Yes

Reviewer #2: Yes

4. Is the manuscript presented in an intelligible fashion and written in standard English?

Reviewer #1: Yes

Reviewer #2: No

Reviewer #2: The idea is not clear

The sample size was 160 and no details on how it was obtained or tge technique used

The concepts arenot clear

Chi2 was mentioned in tge analysis but not clear in the results

All results are figures with no tables

No clear scaling system was used to quantify the data as likert scale

Most of the references used were old

We appreciate the concerns of Reviewer #2 and have modified the manuscript in many places to respond to the various concerns. The reviewer did not provide much guidance in their comments to assist our response but we suggest that the modifications made enhance the paper.

The idea is not clear

We have expanded and modified the introduction section to put our effort in a broader context regarding quantitative education and to point out how it connects to a recent workshop report. 

The sample size was 160 and no details on how it was obtained or tge technique used

We are not sure why the reviewer suggests that there was a sample size of 160. The only place we mention this number is as an approximate number of total graduate students in the various programs, which has no connection to how we carried out our study. There was no sampling of students or interaction with students in these programs associated with this study. The study was based on analysis of a set of 48 journal articles suggested by the faculty. We have modified the manuscript in the Materials and Methods section to emphasize how these papers were chosen based on submissions from the faculty, and we have also attempted to further clarify the objectives of this study in the introduction. 

The concepts arenot clear

We are not certain here as to whether the reviewer is referring to the overall objective and conceptual foundations of the paper or whether the reference is to our use of “concepts” to refer to general quantitative concepts. If it is the former, we have modified the introduction to clarify the objectives further. If the reviewer is referring to our use of the term “concepts” associated with quantitative concepts, we have modified the manuscript to perhaps ameliorate this by consistently using the term “general concepts” and referring to the underlying skills they encompass that are listed in Table S2. 

Chi2 was mentioned in tge analysis but not clear in the results

In the first paragraph of the Statistical data analysis section we have added clarification of the objective of the use of this statistical test. We have also explicitly added mention to these tests in the Results section. 

All results are figures with no tables

The paper does include two Tables that lists the papers and concepts. Presumably the reviewer means data tables that would duplicate the results that we have illustrated in the figures. It is not clear to us how adding tables of these data would enhance comprehension of the paper. As the reviewer responded “yes” to the Question “Have the authors made all data underlying the findings in their manuscript fully available?” we don’t see the necessity of adding these for the purposes of reproducibility. 

No clear scaling system was used to quantify the data as likert scale

We expect that the reviewer is referring here to how the evaluations were assigned by each evaluator to each concept or skill. We have further modified the manuscript to make it even clearer that the assessment of the presence and importance of each concept for each paper was based on the assessment of those from the team with the greatest knowledge of the particular field of the article. There is not a Likert scale here – there is no assumption (as in Likert scaling) of equi-distance between responses along the set of assessments. Rather, these are categorical responses and they cannot be “averaged” in the way Likert scaling might. Our analysis treats the results as discrete categorical values in both the chi-squared tests and in the power analysis we carried out. 

Most of the references used were old

Indeed, we emphasize that though there has been a plethora of recent work on undergraduate quantitative life science education, there has been very little for graduate education. We did add a reference to the recent workshop report on quantitative education that is specifically on graduate life science programs. We added a section in the introduction on this report and how the methods in this paper align with recommendations from that report. We have also added a reference to a recent paper on preprint reviews to enhance education and how this approach can enhance quantitative comprehension through journal clubs.

---

## [Editor Report · Decision Letter 1]

13 Apr 2023

Prioritization of the concepts and skills in quantitative education for graduate students in biomedical science

PONE-D-22-21390R1

Dear Dr. Louis, 

We’re pleased to inform you that your manuscript has been judged scientifically suitable for publication and will be formally accepted for publication once it meets all outstanding technical requirements.

Kind regards,

Yaser Mohammed Al-Worafi

Academic Editor

PLOS ONE
---

## [Editor Report · Acceptance letter]

18 Apr 2023

PONE-D-22-21390R1 

Prioritization of the concepts and skills in quantitative education for graduate students in biomedical science 

Dear Dr. Gross:

I'm pleased to inform you that your manuscript has been deemed suitable for publication in PLOS ONE. Congratulations! Your manuscript is now with our production department. 

Kind regards, 

on behalf of

Professor Yaser Mohammed Al-Worafi 

Academic Editor

PLOS ONE